# Urban Zoning for Sustainable Tourism: A Continuum of Accommodation to Enhance City Resilience

**Chung-Yim Yiu** * and **Ka-Shing Cheung**

Department of Property, The University of Auckland Business School, Auckland 1010, New Zealand; william.cheung@auckland.ac.nz
* Correspondence: edward.yiu@auckland.ac.nz

**Abstract:** While governments around the world are embarking on the path to recovery from the COVID-19 crisis, sustainable tourism planning is crucial, in particular in the hospitality sector, which enhances the resilience of destinations. However, many destination management models overlook the role of urban zoning. Little is known about the impacts of land-use zoning on the hospitality and property industries, especially with the current disruption of short-term peer-to-peer accommodation like Airbnb. Euclidean zoning, also known as effects-based planning, has long been criticised in destination management for its exclusionary nature and lack of flexibility. With exclusionary zoning, property owners may only be able to use their land sub-optimally, and cities will be less efficient in responding to market changes in short-term and long-term accommodation demands, but planning intentions can be better controlled, and the property supply can be more stable. Taking Hong Kong as a noteworthy case, this study puts forward a conceptual framework that enables comparison of a novel zoning approach with the traditional zoning approach. This novel zoning approach encompasses both the short- and long-term rental sectors as a continuum of accommodation, ranging from hotels and serviced apartments to Airbnb and rental housing units under a unified regulatory and planning regime to enhance the switching options value. This novel zoning system can gear up the tourism sector with the rapid growth of the sharing economy and aligns with sustainable tourism to ensure long-term socioeconomic benefits to related stakeholders. We extract the data of Airbnb listings to construct the first Airbnb ADR Index (ADRI) by Repeat-sales method, and the results support our Switching Option Hypothesis.

**Keywords:** sustainable destination management; city resilience; exclusionary zoning; accommodation continuum; sharing economy

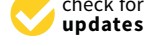

## 1. Introduction

The peer-to-peer market, collectively known as the collaborative economy, has emerged as an alternative supply of accommodation that blurs the boundaries between the traditional hotel industry and residential rental housing [1]. First, hotels are traditionally considered to have commercial purposes, and are governed by different sets of regulations and tax policies compared to residential rental housing. However, the emergence of peer-to-peer markets such as Airbnb has made the difference unclear. Even owner-occupied housing units can easily earn rental income by subletting either bedrooms or the whole unit for short periods to visitors.

Second, while hotels are designed for short-term accommodation, residential rental housing is for the long-term [2]. Indeed, the difference between short- and long-term accommodation also becomes ambiguous, and their definitions can be arbitrary. Taking Hong Kong as an example, on the one hand, the Hotel and Guesthouse Accommodation Ordinance of the Laws of Hong Kong (Cap. 349) [3] defines a hotel or guesthouse as a premise providing sleeping accommodation for fewer than 28 consecutive days. Any persons who want to stay for longer than 28 days on a premise are required to rent a

residential unit. On the other hand, the Landlord and Tenant (Consolidated) Ordinance in Hong Kong governs only long-term rental housing units. The emergence of service apartments and Airbnb makes the rules challenging to enforce.

Third, hotels are built on commercial land-use zones, and housing units on residential land-use zones, with very different planning and building regulations. The primary purpose of land-use zoning is to separate incompatible land uses so as to avoid causing a nuisance, but short-term and long-term accommodation purposes are similar and compatible uses. When short- and long-term accommodation uses are designated as different uses in town planning, more land is required to satisfy the peak demands of accommodation from both visitors and residents and this can result in a more serious spatial mismatch and higher land prices.

The demands of short- and long-term accommodation can exhibit very different patterns. While short-term accommodation depends substantially on seasonal events that attract tourist arrivals, long-term accommodation hinges on the growth of the population. A complete segmentation between these two distinct types of properties will require a much larger contingency in each market to cater for the peak demands. In the low season there will be a lot of vacant accommodation. A large city with an ample supply of land and properties may not have an issue, but in small, high density cities with a limited land supply (such as Hong Kong), a severe spillover problem is possible. Economic theory predicts that some market participants will take the risk of non-compliance if the extra return can compensate the risk [4,5]. During overtourism, the insufficiency of hotels and guesthouses in a destination places short-term accommodation such as Airbnb in high demand. Residential property owners will have an increased incentive to make their properties available for short-term housing.

Even though Airbnb is not legally allowed in some cities, including Hong Kong [6–9], the popularity of Airbnb in these cities reveals the difficulty in enforcing the controls. For example, the data from Inside Airbnb [10] indicates that in January 2019 23,220 active Airbnb units were looking for occupiers in Hong Kong, a figure accounting for almost 2% of the total private housing stock. Other city governments accept the existence of Airbnb with or without charging commercial taxes, which implies that the zoning separation between short-term and long-term accommodation has become, de-facto, non-existent.

One of the major criticisms of Airbnb is its inflationary effects on rents as the supply of housing units is allegedly switched from long-term to short-term accommodation. However, almost all the supporting empirical evidence is manifested in overtourism periods. During the COVID-19 pandemic, there was a substantial fall in housing rents in Hong Kong. Is the Switching Option Hypothesis valid in the downturn? Since the pandemic outbreak in December 2019, many cities have prohibited the entry of visitors, and this has abruptly turned the tide from overtourism to undertourism. The occupancy rates of hotels and short-term accommodation have plummeted to an unprecedented low. For example, house rents in Hong Kong fell by 7.3% in 2020 (Figure 1). The number of active Airbnb units fell to 6986 units in January 2021 [10], a 70% drop compared to the same month in 2019. Although there are no statistics to show how many housing units have been switched from Airbnb back to the long-term rental market, anecdotal evidence suggests that landlords will simultaneously list their properties on two markets because the switching cost is negligible.

The flexibility in switching land uses may cause volatility in housing rent, but it also enables the city to receive more visitors to provide more job opportunities. The World Tourism Organisation defines sustainable tourism as "tourism (that) takes full account of its current and future economic, social and environmental impacts, addressing the needs of visitors, the industry, the environment and host communities" [11]. The principles of sustainable tourism emphasise a viable tourism development strategy that will "respect the socio-cultural authenticity of host communities, . . . and provide . . . stable employment and income-earning opportunities and social services to host communities." How to balance these sustainable tourism goals is challenging. To discuss the costs and benefits of mixed

use between the residential and hospitality sectors, this paper aims to put forward a novel concept of a continuum of accommodation.

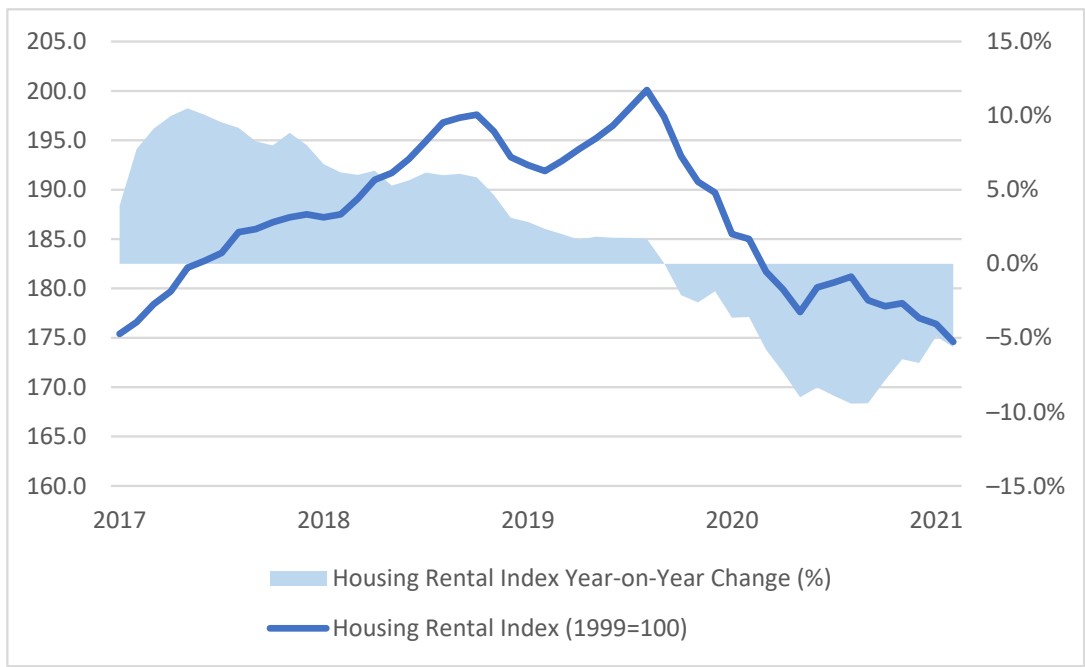

**Figure 1.** Hong Kong Housing Rental Index and its growth, January 2017–Feburary 2021. Source: the chart is constructed by the authors based on the data collected at [12] Notes: The private housing rental index of Hong Kong started to fall in October 2019, which was earlier than the WHO announcement of the outbreak of the COVID-19. The market downtrend probably reflected the impacts of the social unrest in 2019.

## 2. Literature Review

Within the urban planning literature, the distinction between regulatory planning versus strategic planning is related to a long-standing debate on a recurring dichotomy between notions of a 'conforming' system, which refers to the regulative certainty through zoning, and a 'performing' system, which emphasises strategic flexibility through discretionary planning [13]. Experiences in various cities have demonstrated that regulatory zoning rarely delivers certainty and often exercises discretion to ensure a city's sustainability and economic growth [14]. Regulatory zoning helps shape the built environment by dividing communities into distinct urban zones, and determines what 'can' or 'can not' be built within those discrete zones. The core purpose of regulatory zoning is to create certainty by removing and separating conflicting land uses and dysfunctional districts that may impede investments in the land [15]. Therefore, the structure of regulatory zoning provides a powerful tool for directing changes through land-use controls. Carmona et al. [16] advocated that regulatory zoning is an economic tool to protect property and land values. This zoning approach can also be deployed as a social tool to preserve public space and amenities from those unacceptable development activities [17] and as an environmental tool seeking to regulate the impacts of pollution and by segregating potentially noxious industrial uses [18].

However, the reality is that growing flexibility with the use of discretion based on subjective interpretation by planners and other stakeholders continues to prevail. Entrepreneurs, architects, and developers have interpreted regulatory zoning not as a restriction, but rather as a form-giving principle. Regulatory zoning is considered to be essentially a technical and professional exercise that schematises rational land-use planning [19]. Consequently, regulatory zoning is constantly being criticised as a static land-use tool that cannot evolve to keep pace with the complex changes in reshaping urban forms. Booth [20]

even suggests that all zoning systems will sooner or later result in a mismatch between the predicted future and the realities of the moment. All zoning systems develop mechanisms for circumventing the rules that were supposed to set the limits for future action.

The contemporary context of land-use practices in most developed countries is to create more resilient planning that can embrace the growing uncertainties in urban development. The planning ambition is to balance the certainty (efficiency) of land-use systems and planning flexibility to address the rapid urbanisation. Indeed, many jurisdictions demonstrate a combined use of the regulatory zoning and discretionary regimes. In the United States, 'hybrid planning' has evolved in an attempt to embrace both flexibility and certainty like 'flexible zoning' [19] and the United Kingdom has 'planned discretionary systems' [16]. These changes have been facilitated by a performance-based planning emphasis with contemporary understandings including from the level of strategic spatial policy [21,22], and the use of performance indicators and codes at the level of local planning schemes [19,23]. Rivolin [13] argues that the distinction between the conforming (regulatory) and performing (flexible) planning systems is the modality for delivering development rights within an urban planning framework. In 'conforming' systems, the rights are assigned in advance and translated into zoning ordinances; in the 'performing' systems, however, development rights are designated as a non-binding policy reference that must be assessed in conjunction with the strategic framework [24,25].

With the emergence of 'collaborative economies' such as Airbnb, it is crucial to have performing planning. Airbnb is one of the leading business models of the collaborative economy, and is based on a worldwide peer-to-peer short-term online rental platform. Many studies have been conducted to investigate how the Airbnb business model affects the hospitality industry [26–28], but Guttentag [29] contended that there is no study on how to position Airbnb to achieve the principles of sustainable tourism.

Economists also recognise that the Airbnb market enables new demand for residential investment beyond traditional urban markets [30]. Lee [31] argues that Airbnb reduces the housing supply in Los Angeles and increases housing rents. Subsequently, there have been several empirical studies of the impacts of Airbnb on housing rents. For instance, Horn and Merante [32] showed that one standard deviation of increase in Airbnb listings is associated with a 0.4% increase in asking rents in Boston. Franco et al. [33] and Barron et al. [34] found that a 1% increase in Airbnb listings is associated with 2.26% and 0.018% increases in rents in Portugal and the United States, respectively. Studies also indicate that cities and regions affected by rental shortages and strong demand for tourist accommodation are particularly susceptible to the impacts of Airbnb-like platforms [35–37]. When there is a tourism demand for residential-style accommodation, small cities may struggle to sustain affordable housing [38–40].

However, the popularity of Airbnb reflects its benefits to stakeholders. For example, it is welcomed by both short-term occupiers and property owners for cost-saving and profit-making. Airbnb is also said to create new job opportunities [41] (*"Airbnb community generates $824 million in economic activity in the UK, supports 11,629 Jobs, helps residents meet cost of living challenges".*) and reduce crime [42]. Cheung and Yiu [43] found that Airbnb can have either a positive or a negative association with housing rents depending on the compatibility of the short-term and long-term accommodation environments. This finding implies that the impacts of Airbnb on neighbouring housing markets rely on land-use zoning factors.

Regan et al. [44] posit that considering option values will provide a more realistic assessment of landholder investment decision making. Switching options is one of the five standard options considered in real options analysis, but the previous studies of switching options are usually concerned with switching among markets, technologies and products, rather than switching between land uses. Studies of switching options of land use are commonly focused on switching agricultural crops and on conservation [45–48], and very few of them have analysed the possibilities of the switching of land-use options allowed by zoning laws.

Other applications of real options analysis in real estate include redevelopment options [49,50], development-intensity options [51] and bidding strategy options [52]. There have been some studies of mixed land-use option values that provide theoretical and empirical supports to our conceptual framework, but most of them consider land-use choice in redevelopment rather than switching between long-term and short-term accommodation. For example, Capozza and Li [51] dealt with the irreversible conversion from one use to another. Geltner, Riddiough and Stojanovich [53], p. 20, extended the previous studies to allow mixed uses and found that "land-use choice (i.e., multiple-use zoning) may add over 40% to land value". A major reason for the value of the switching option is the differences in market demands for, and supplies of different land uses at different time. Childs, Riddiough and Triantis [54], p. 319, extended both models by allowing multiple conversions between uses and concluded that "in the presence of relatively low costs to conversion, flexibility with respect to mixing uses and redevelopment will contribute significantly to the overall value of the built property". Their studies established the value of providing flexibility in land uses to the property owners. Yet, in this strand of studies, there are two research gaps. First, they do not consider social and environmental planning costs, but instead focus only on economic benefits to the landowners. Second, their options usually refer to the conversion of land uses by redevelopment, which is either irreversible or very expensive to reverse. In contrast, switching between long-term and short-term accommodation is highly reversible and seemingly cost-free.

This paper aims to fill the two research gaps by putting forward a conceptual framework, from an option pricing perspective, on the relationships between Airbnb and housing rental units. The sharing economy enables a single-use house to have an option to switch fully or partially between residential or hospitality land use at seemingly no cost to the property owners. This increases land value, and it is an option that housing owners can choose to exercise or not to.

## 3. Materials and Methods

### 3.1. Switching Options Embedded in Performing Planning

Conceptually, the effects of Airbnb on the housing rental market and hence residential land-use planning can be simplified by considering a city of only four square parcels of land. We compare the scenarios of a City A with Airbnb and a City B without Airbnb.

In the initial state of the four squares of land uses in City A and City B, each of the three uses of land—housing, hotel and others—occupies one square each. This leaves one blank square as vacant land for future development. Suppose both cities are facing a very high demand for tourist accommodation. In that case, the number of hotel rooms in City A becomes insufficient, and the lack of Airbnb prevents the city from accommodating visitors' demand. In the longer run, if the overtourism situation is expected to be continuous, then vacant land will be developed for hotel use. In contrast, City B will accommodate more visitors, as the homeowners can switch their homes to become Airbnb. However, in that case, the reduction in housing supply for long-term rental accommodation may cause higher housing rents, at least in the short run. Thus, the vacant land is more likely to be developed into residential use (Figure 2a,b).

In other words, in equilibrium, Airbnb does not reduce the supply of rental housing if there is land to build on. The trade-off is a short-term increase in housing rents or a cap on the number of visitors during an overtourism scenario. This shows that mixed land use for both residential and hospitality provides flexibility in accommodating short-term volatility in demand, but the cost can be higher volatility in rents.

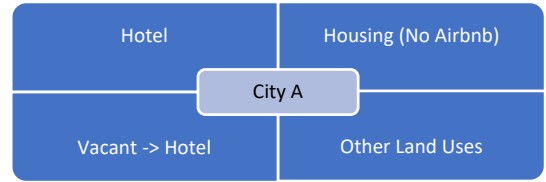
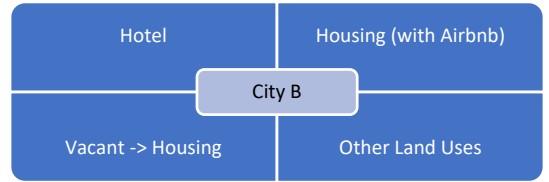

(**a**) Hotel land use of the vacant land in City A (without Airbnb)

(**b**) Housing land use of the vacant land in City B (with Airbnb)

**Figure 2.** Land use of vacant land in two city scenarios with and without Airbnb. (**a**) refers to the scenario without Airbnb, (**b**) refers to the scenario with Airbnb.

In contrast, during undertourism periods, City A will encounter a situation of excessively high vacancy rates for hotel rooms, but City B will be more resilient in absorbing the fall in the number of visitors as Airbnb can be efficiently switched back to long-term rental use. The option for homeowners to switch uses for homeowners demonstrates the option value during the pandemic, and they help to share the losses from undertourism with the hotel sector. The basic premise of the conceptual framework is that mixed land uses can help reduce vacancy rates by providing flexibility. It can therefore reduce deadweight loss and save land supply. This is a novel analytical idea of the sharing economy, and is drawn from a town planning perspective. While Airbnb is commonly analysed from the home space sharing perspective, this paper explores Airbnb from the land-use sharing view by putting forward an option pricing concept.

Similarly, during the pandemic, most hotels have been switched to accommodate quarantine stays. This helps hotels to reduce vacancies and protects the city from making the spread of the virus worse. Undoubtedly, a positive outcome is achieved through exercise of the option to switch the building's use from a hotel to a quarantine centre. There are hardly any available facilities in the cities that can be mobilised within such a short time to accommodate inbound persons for quarantine purpose. Hotels are good examples of switching to an urgent use at low cost and keeping the hotel operators in business.

The novelty in the proposed zoning approach relies on the flexibility of switching land uses between hotel, service apartment, peer-to-peer accommodation and housing rental. All these uses can be categorised as Accommodation Uses to maximise the value of the switching option and at the same time keep the land uses conflict to a minimum. Since all these uses are for accommodation, they share a highly similar requirement on facilities and environment, especially when work-from-home and e-shopping have become more common after the outbreak of COVID-19.

In fact, areas with all these four uses in small neighbouring zones can easily be found in Hong Kong, indicating that they are compatible in land use and sharing the facilities in the neighbourhood, but not switchable freely. For example, Figure 3 shows a current zoning plan of an area of about 17 km². Hotels can be built on all Commercial/Residential land and some Other Specified Uses (Commercial Development with Piblic Transport Interchange) land (blue- and purple-coloured areas), whereas housing units can be built on Residential land (dark-green-coloured areas) and Commercial/Residential land (blue-coloured areas) [55]. The compatibility between short-term and long-term accommodation is revealed in this example where commercial zone and residential zone are located next to each other or even in the same zone.

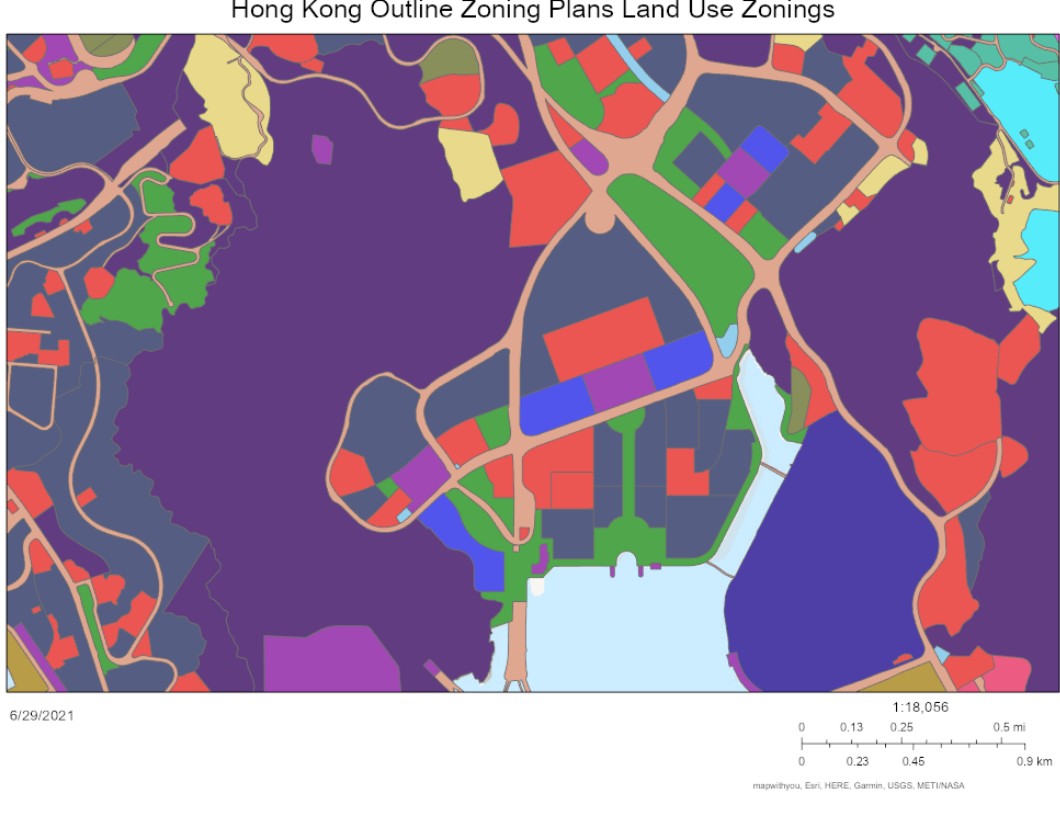

**Figure 3.** Outline Zoning Plan (OZP) in Tseung Kwan O Area (No. S/TKO/28) Reprinted with permission from ref. [55]. Copyright © 1995–2020 Esri. Available online: https://www.arcgis.com/apps/mapviewer/index.html?webmap=5375a8 8ec76143ea974d5fb64efbec0d (accessed on 29 June 2021).

### 3.2. A Case Study of Serviced Apartments in Hong Kong

The option to switch land use is not new for hotels. For example, the timeshare of resorts and vacation homes has been developed in the 1960s. There was a case in point in Hong Kong, which showed the effects of town planning controls on the option to switch hotel use. Before the year 2000, the serviced apartment was considered as a hotel use by the Town Planning Authority, but it was considered a residential use by the Building Authority. Many serviced apartments were then built on hotel zones but were sold to individual buyers in strata-title. This option to switch between hotels and serviced apartments provided developers with the flexibility to maximise profits when the developments were ready to sell. However, in 2000, the Town Planning Authority set up a task force to look into the issue. After a one-year study, the task force concluded that serviced apartments could not easily and legally be differentiated from hotel rooms and housing units. The Authority issued a practice note (TPB PG-NO. 2B) [56] in June 2000 to stop the strata-title sale of serviced apartments and thereby prevented the development of conventional residential flats in the name of serviced apartment in inappropriate areas.

The reason for the decision has been claimed to be "land use incompatibility and problems of inadequate provision of supporting community facilities" [56], as land for short-term accommodation can be zoned in industrial and commercial areas or areas subject to environmental constraints. However, in reality, many residents and expats live in serviced apartments for a relatively long period. This case illustrates the trade-off between land uses as long-term and short-term accommodation. The emergence of Airbnb provides

short-term occupiers with an option to choose either hotels, serviced apartments or Airbnb, but long-term renters cannot stay at hotels or serviced apartments unless they move out at least one day for every consecutive 28 days.

This case shows a real-life experiment of allowing hotels and serviced apartments to be sold in strata-title as normal housing units. It makes better use of the land resources and helps to dampen increasing residential land price. Yet, it undermines the original intentions of the land-use zoning system. Serviced apartments and Airbnb can be considered as the two mirror images of the switching option. Airbnb allows homeowners to switch a housing unit from residential use to hospitality use, whereas serviced apartments provide hotel owners with the options to switch from hospitality use to residential use (Figure 4).

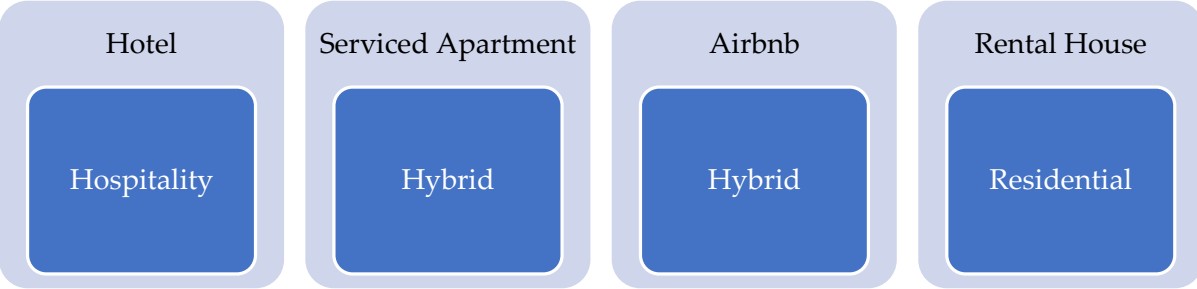

**Figure 4.** The continuum between hospitality and residential properties.

### 3.3. Repeat-Sales Method

To study the option value in switching between short-term and long-term accommodation markets, we have to identify the association between the market rents of the two markets, all other things being equal. However, so far, there is no quality-adjusted rental index of short-term accommodation available for comparison. This study uses the data of Airbnb listings and ADR of Hong Kong from April 2018 to February 2021 from *Inside Airbnb* to construct the first Airbnb ADR Index using the repeat-sales method.

By design, the repeat-sales method [57] constructs the price index by comparing the price difference between a pair of repeated sales of the same house, so it does not require the data of temporal-invariant attributes. It is, therefore, an appropriate tool for estimating the ADR index of Airbnb, as the listings usually lack most of the housing attributes and are updated only monthly with their ADRs.

The repeat-sales ADR index (RSADRI), as shown in Table 1, assumes that the property characteristics and implicit prices of the same property are invariant between the first sale ($t_1$) and the second sale ($t_2$), and the following equation shows how the repeat-sales model estimates the monthly index; Equation (1):

$$ln\left(\frac{P_{it_2}}{P_{it_1}}\right) = \sum_{t=1}^{T} \alpha_t D_{it} + \varepsilon_{it_1 t_2} \tag{1}$$

where $P_{it_1}$ and $P_{it_2}$ denote the ADRs of property $i$ in period $t_1$ and $t_2$ ($i = 1, \ldots, n$); $\alpha_t$ denotes the estimated coefficient for time dummy $D_{it}$ ($t = 0, \ldots, T$), and $\varepsilon_{it1t2}$ is the error term. The time dummies are set to take the value $-1$ if $t = t_1$, $+1$ if $t = t_2$, and zero otherwise. To normalise the price index at time period $t = 0$, $\alpha_0$ has been set to zero so as.

**Table 1.** The estimated repeat-sales Airbnb Average Daily Rate Index (RSADRI) for Hong Kong, April 2018–February 2021.

| yyyymm | RSADRI (201804 = 1) | yyyymm | RSADRI (201804 = 1) | yyyymm | RSADRI (201804 = 1) | yyyymm | RSADRI (201804 = 1) |
|---|---|---|---|---|---|---|---|
| | | 201901 | 1.0290 | 202001 | 0.9913 | 202101 | 0.9674 |
| | | 201902 | 1.0294 | 202002 | 0.9885 | 202102 | 0.9711 |
| | | 201903 | 1.0304 | 202003 | 0.9858 | | |
| 201804 | 1.0000 | 201904 | 1.0352 | 202004 | 0.9845 | | |
| 201805 | 1.0020 | 201905 | 1.0216 | 202005 | 0.9855 | | |
| | | 201906 | 1.0196 | 202006 | 0.9859 | | |
| 201807 | 0.9997 | 201907 | 1.0144 | | | | |
| 201808 | 0.9996 | 201908 | 1.0101 | | | | |
| 201809 | 1.0068 | 201909 | 1.0055 | | | | |
| 201810 | 1.0139 | 201910 | 1.0015 | 202010 | 0.9562 | | |
| 201811 | 1.0204 | 201911 | 0.9979 | 202011 | 0.9600 | | |
| 201812 | 1.0264 | 201912 | 0.9951 | 202012 | 0.9637 | | |

## 4. Results and Discussion

### 4.1. Results

In a small city like Hong Kong, where land supply is scarce, different land-use designations for short-term and long-term accommodations could result in high land prices. The lack of flexibility could also cause a more serious waste of resources during a market downturn. For example, during the outbreak of COVID-19, the average annual vacancy rate of hotels increases to about 55%, as compared to a vacancy rate of just 30% during SARS in 2003. In contrast, the average annual vacancy rate of housing was just 4.3% in 2020, which is close to the natural vacancy rate (Figure 5)

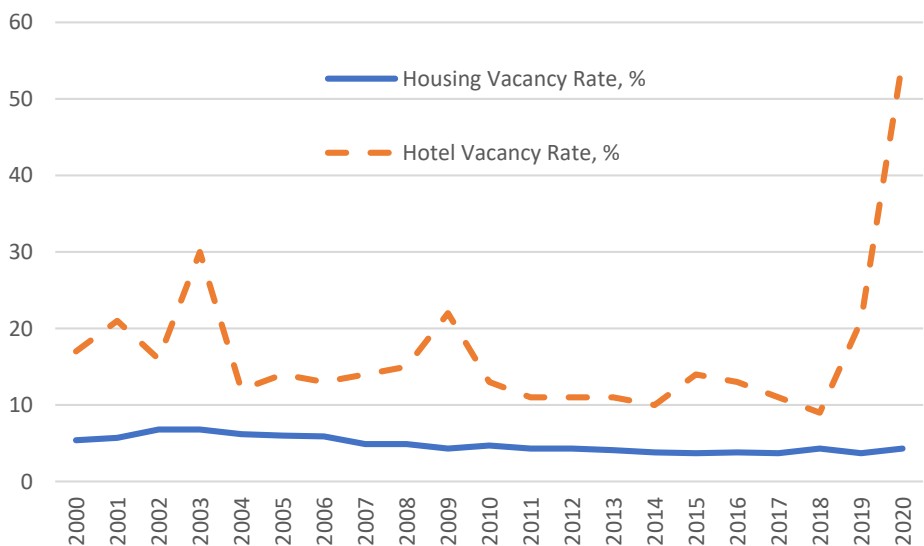

**Figure 5.** Housing vacancy rate and hotel vacancy rate of Hong Kong, 2000–2020. Sources: the chart is constructed by the authors based on the data collected at [12,58]. Notes: The hotel vacancy rate is calculated by subtracting the reported hotel occupancy rate from one each year [58]. The annual average vacancy rate of private housing is directly extracted from [12].

In contrast, Airbnb can mitigate the negative impacts of shocks by either reducing its rent—Average Daily Rate (ADR)—or by switching the accommodation to the long-term rental market. We extract the Hong Kong data on Airbnb listings and ADR from April 2018 to February 2021 from *Inside Airbnb* to construct the Airbnb ADR Index (ADRI) by the repeat-sales method. Figure 6 shows that as the number of visitors decreased gradually from early 2019, the ADR followed the downward trend closely, but the number of Airbnb

listings was kept on the highest plateau of more than 12,000 listings per month until the outbreak of COVID-19 in early 2020. It indicates that in this pre-COVID period, the property owners tried to cut the ADR without switching to the long-term rental market. However, the pandemic caused a sudden plummet in visitors from more than three million per month to almost zero. Both the ADR and the number of listings of Airbnb fell together to unprecedented lows. In other words, after the outbreak of COVID-19, some property owners further cut the daily rents, while others stopped listing their properties for short-term accommodation. Some of these owners likely switched to the long-term rental markets and reduced their housing rents.

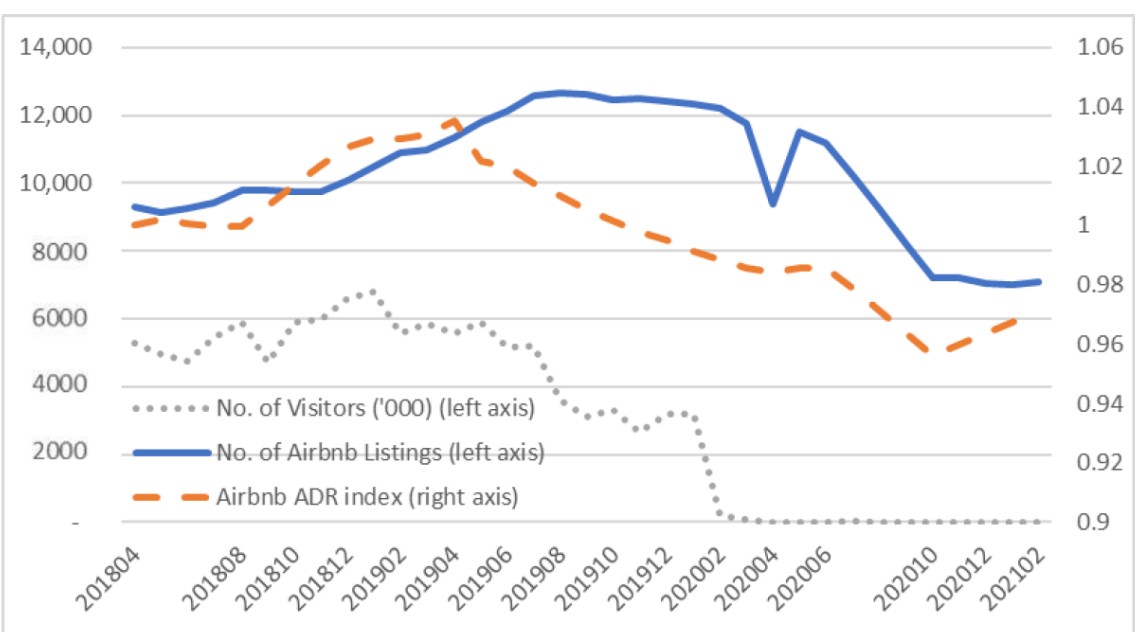

**Figure 6.** Airbnb's Average Daily Rate (ADR) index and number of Airbnb Listings of Hong Kong, April 2018–February 2021. Sources of data: the chart is constructed by the authors based on the data collected at [10,58]. Notes: The monthly total number of visitors data is directly extracted from [58], the Airbnb listings data are taken from [10], and the Airbnb ADR index is estimated and constructed by the authors using the repeat-sales method on the data provided by [10]; details of the estimation method and the index are provided in Section 3. Please note that the data in June 2018 and from July 2020 to September 2020 are unavailable at *Inside Airbnb*. The authors interpolate the corresponding figures.

We further compare the ADR of Airbnb and the market rental index of private housing in Hong Kong to show the association between short-term and long-term rents in the city. Figure 7 shows the annual growth rates of ADR and housing rents from April 2019 to February 2021. Both rates have become negative since September/October 2019, and the two growth rates track closely with each other with a positive correlation coefficient of 87.3%. With the outbreak of COVID-19, private housing market rents fell more substantially than the ADR of Airbnb. On the one hand, this probably reflects the sharp reduction in the demand for long-term rental accommodation when international students and ex-pats could not come. On the other hand, switching properties from short-term offers to long-term rental offers further increases the market supply. The data agree with the Switching Option hypothesis.

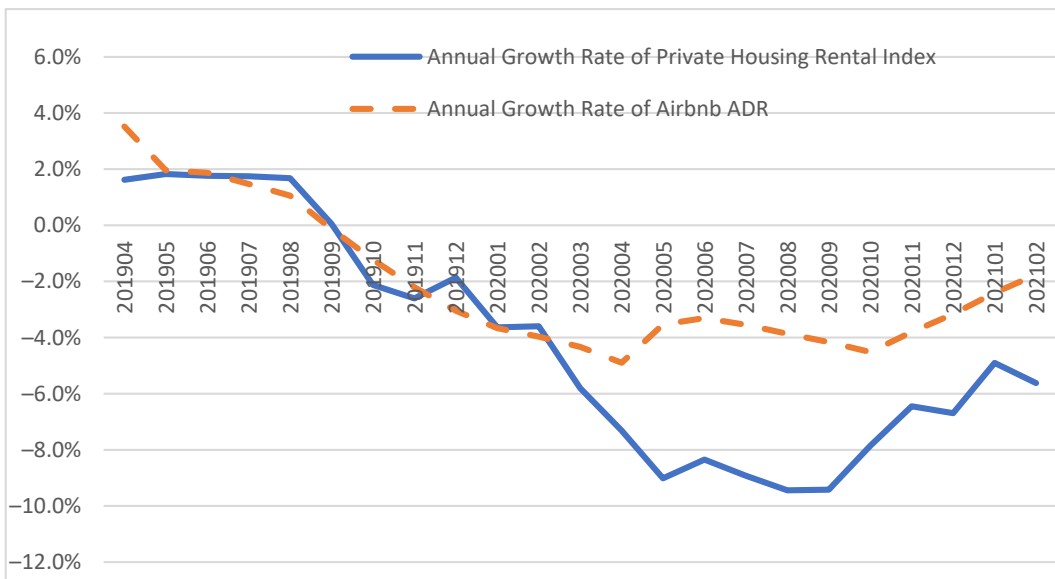

**Figure 7.** Annual growth rate of Airbnb's Average Daily Rate (ADR) index and annual growth rate of private housing rental index in Hong Kong, April 2019–February 2021. Sources: the chart is constructed by the authors based on the data collected at [10,12]. Notes: The annual growth rate of Airbnb ADR data is estimated and constructed by the authors using the repeat-sales method on the data provided by *Inside Airbnb* [10]. Details and the index are provided in Section 3. The annual growth rate of the Private Housing Rental Index is estimated from the data provided by [12].

In view of the low vacancy rate of private housing and the highly unaffordable housing rents in Hong Kong, exercising switching options from short-term to long-term accommodation during undertourism periods can help make housing rents more affordable. The results also reflect that the switching option provides flexibility for property owners to mitigate vacancy risk in both over- and under-tourism periods. The benefit of multiple land use flexibility is particularly important when the demands of short- and long-term accommodations are volatile. However, the case study also reveals that the switching option can exacerbate short-term housing rent volatility.

*4.2. Discussion*

So far, most studies [31–40] on short-term and long-term accommodation have considered them as two distinct types of properties in order to investigate the impacts of either one on the other, without realising that they are actually the same group of properties serving the same purpose of accommodation. These two types of properties can be switched from one to the other at almost zero marginal costs if regulations allow. More recently, with the enactment of the Home Sharing Ordinance in the US, it was found that short-term rental units were switched back to the long-term rental market [2]. Yet, they focused on the efficient use effect, rental housing supply effect and the externality effect, without analysing the switching option. The switching option hypothesis in short- and long-term accommodation is an almost unexplored area in the literature, which justifies further research.

Furthermore, previous studies have usually examined the effect of the increase in Airbnb listings on housing rents rather than the change in ADR. In their rental housing supply hypothesis, the relationship between the ADR and the supply demand of Airbnb is largely ignored. This is probably because of the lack of an accurate ADR index for analysis. This study, therefore, puts forward a repeat-sales method to construct an RSADRI index. This study, using the index, shows a positive association between short-term and long-term accommodation rents in Hong Kong, but not between the number of Airbnb listings and long-term housing rents in 2019. It raises a challenge to the rental housing supply hypothesis and supports the switching option hypothesis.

However, as this study aims to put forward a conceptual framework of a novel zoning approach that encompasses both short- and long-term rental sectors as a continuum of accommodation, ranging from hotels, service apartments to Airbnb and rental housing units under unified regulation and planning to enhance the switching options value, we have not conducted rigorous empirical analyses on the hypotheses. It requires further empirical studies to examine the hypotheses.

## 5. Conclusions

Regulatory land-use planning is generally known as 'non-performing', and the time frame for the bureaucratic decision-making on rezoning are usually too long. Inadequate public involvement and dissatisfaction with planning outcomes are common [59]. Planning systems worldwide have attempted different hybrid forms to optimise the flexibility and certainty of urban developments. This includes the use of flexible zoning–the exercising of discretionary planning, and the promotion of non-binding planning [15,18,60,61]. The shift emphasises how land-use planning can 'perform' on selective strategic and action-oriented visions rather than 'conform' to technical, legal regulation, or land-use plans [61,62].

As our cases demonstrated, a rigid differentiation between short-term and long-term accommodation is not only difficult to enforce, but also limits flexibility in responding to shocks to market demands. On the one hand, property owners lose their switching option values and bear the vacancy cost in a market downturn. On the other hand, the city as a whole will suffer a deadweight loss through the inefficient use of land resources and will have to provide more land to accommodate the various uses. Rivolin (2008) (p. 171, [13]) advocated that the crux for urban planning system is not whether conforming or performing plans are preferable but rather "how the strategic and regulative functions of planning should be differently correlated in a planning system addressed to performing rather than conforming aims". However, the dichotomy between rigid regulatory zoning as a means to achieve certainty and strategic discretionary systems pursuant to achieving flexibility as a defining feature differentiating planning systems continues to limit our understanding of choosing either approach. Indeed, rather than a dichotomy between strategic (discretionary) and/or regulatory (zoning) planning systems, the focus should be in the performance of planning strategies as a whole, incorporating both top-down and bottom-up processes. A principle of flexible planning and development is to promote a commitment to sustainability and to reduce red tape in order to enhance the efficiency of regulatory planning. The recurring discussion on performance-based planning reinforces the need to move away from fragmented and costly development control mechanisms, cumbersome impact assessment procedures, and an over-reliance on rezoning, toward a more flexible, creative, and ultimately market-driven planning alternative.

The flexibility versus certainty debates somehow distract scholarly attention away from the substantive issues of how urban planning institutions can better create sustainability in the era of the collaborative economy. The ongoing debate of balancing certainty with flexibility in planning systems conceals the genuine concerns of embracing the challenges in the planning systems. Planning and legislative reform need to slash red tape and adopt a more holistic and integrated vision of urban governance. Such a concept is more likely to be achieved through context-specific and hybrid legislative arrangements. The provision of switching options can undermine the original planning intentions and cause a short-term increase in rents during a market upturn.

The case of selling serviced apartments as residential properties in Hong Kong shows how market forces encourage the exploitation of option switching. The government stopped such sales by an amendment to the Outline Zoning Plan, but the recent emergence of Airbnb is another attempt of the individuals responding to market changes to exercise switching options, and it is hard to stop by regulations this time. The COVID-19 pandemic clarifies that switching options can help a city better respond to shocks. The benefit of multiple land-uses is particularly valuable when the demands of short- and long-term

accommodations are volatile, as predicted by the option pricing theory. The authors intend to extend the perspectives in this conceptual paper by means of empirical tests.

**Author Contributions:** Conceptualisation, C.-Y.Y. and K.-S.C.; formal analysis, C.-Y.Y. and K.-S.C.; investigation, C.-Y.Y.; data curation, K.-S.C.; writing—original draft preparation, C.-Y.Y.; writing—review and editing, K.-S.C.; funding acquisition, C.-Y.Y. Both authors have read and agreed to the published version of the manuscript.

**Funding:** The APC was funded by the University of Auckland research grant number 1590/50036.

**Institutional Review Board Statement:** Not applicable.

**Informed Consent Statement:** Not applicable.

**Data Availability Statement:** Publicly available datasets were analysed in this study. These data can be found in the following links: [10] Inside Airbnb. Detailed Listings data for Hong Kong, *Get the Data*. 2021. Available online: http://insideairbnb.com/get-the-data.html (accessed on 14 April 2021); [12] RVD. *Private Domestic–Rental Indices by Class (Territory-wide) data series*, Rating and Valuation Department, HKSAR Government. 2021. Available online: https://www.rvd.gov.hk/mobile/en/property_market_statistics/index.html (accessed on 14 April 2021); Ref. [58] Hong Kong Tourism Board. *Research & Statistics*, Hong Kong Tourism Board. 2021. Available online: https://partnernet.hktb.com/en/research_statistics/tourism_statistics_database/index.html (accessed on 14 April 2021). The estimated ADR index of Airbnb in Hong Kong from April 2018 to February 2021 is available in Section 3.

**Conflicts of Interest:** The authors declare no conflict of interest.

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
