# Peer review of "Urban Zoning for Sustainable Tourism: A Continuum of Accommodation to Enhance City Resilience"

_sustainability, doi:10.3390/su13137317_

Round 1

Reviewer 1 Report

Review comments: Manuscript ID:- sustainability-1251148

General Comment

This is an interesting manuscript urban zoning for sustainable tourism in the study region. However, there are aspects that require improvement and clear presentation before being considering for publication. Specific comments and suggestions are included below.

Specific comments and suggestions

L7-22: I suggest the authors including brief description of datasets used, major findings and conclusions in the abstract.

L26-28: Add source(s)

L34-35: Add source(s)

L36-38: Add source(s)

L204-352: Authors need to properly describe their used approach as “material and method” section and their findings as “results” section. The current version mix these both section and create confusion to the results.

L353: Before “concussion” section, I also suggest a discussion section. The major findings need to discussed thoroughly. I suggest authors to reflect the following information during discussion: how this study will increase our knowledge base and inspire others to conduct further research, how the   results (not) support findings of earlier studies, discuss the major results in the context of answering the formulated research objective, whether your findings agree with current knowledge and expectations, and any weaknesses in the results/approaches and suggest room for further research concerning that aspect of your analysis.

Reviewer 2 Report

The authors of the article justify the need of flexible multi-functional zoning of urban areas to enhance city resilience during the periods of quickly changing accommodation possibilities due to pandemic situation and possible other reasons of “force majeure” type. The idea is not new. All the world moved from rigid mono-functional zoning to the more flexible multi-functional zoning in the second half of the 20th century. The novelty of the research is a prove of effective functional mix of short-time and long-time rental sectors. The research material is well-structured, systematized and logically and coherently presented. The title of the article reflects the content of the research. The abstract and keywords are appropriate. Introduction encompasses the description of the research relevance and the aim of the research. The main text represents literature review on the topic and case study results. Conclusions summarize the main findings of the research. List of references is appropriate.

Reviewer 3 Report

Urban Zoning for Sustainable Tourism: A Continuum of Accommodation to Enhance City Resilience

This article focuses on a novel zoning approach for Sustainable Tourism.

The study puts forward a conceptual framework to compare a novel zoning approach with the traditional zoning approach (line 17-20).

They state that the traditional regulatory zoning is constantly being criticised as a static land-use tool that cannot evolve with the complex changes in reshaping urban forms (line 124-126).

Reading the paper, it’s not clear the novelty in the proposed zoning approach.

Authors have to introduce a specific paragrapf in which they explain their proposal of the new zoning.

e.g.:

are there some range of the function rates (land use percentage) in each specific zone?

what about required facilities in each zone? They do have to depend on the specific function (land use) present in the specific zone?

Could be useful provide a schematic map of Hong Kong axplaining the prevalent land use.

Round 2

Reviewer 1 Report

Review comments: Manuscript ID:- sustainability-1251148

The manuscript (ID: sustainability-1251148) entitled with “Urban Zoning for Sustainable Tourism: A Continuum of Accommodation to Enhance City Resilience” has gone through a significant revision as compared to the earlier version. Major issues from my side were already taken into account.

The manuscript has merit for optimizing urban land use by forwarding a conceptual framework that enables comparison of a novel zoning approach with the traditional zoning approach. The research is based on a case study. Such kind of study is important for providing an option that owners can choose.